# Microthermal-induced subcellular-targeted protein damage in cells on plasmonic nanosilver-modified surfaces evokes a two-phase HSP-p97/VCP response

Martin Mistrik[1,6 ✉], Zdenek Skrott [1,6], Petr Muller [2], Ales Panacek [3], Lucie Hochvaldova [3], Katarina Chroma[1], Tereza Buchtova[1], Veronika Vandova [2], Libor Kvitek [3] & Jiri Bartek [1,4,5 ✉]

Despite proteotoxic stress and heat shock being implicated in diverse pathologies, currently no methodology to inflict defined, subcellular thermal damage exists. Here, we present such a single-cell method compatible with laser-scanning microscopes, adopting the plasmon resonance principle. Dose-defined heat causes protein damage in subcellular compartments, rapid heat-shock chaperone recruitment, and ensuing engagement of the ubiquitin–proteasome system, providing unprecedented insights into the spatiotemporal response to thermal damage relevant for degenerative diseases, with broad applicability in biomedicine. Using this versatile method, we discover that HSP70 chaperone and its interactors are recruited to sites of thermally damaged proteins within seconds, and we report here mechanistically important determinants of such HSP70 recruitment. Finally, we demonstrate a so-far unsuspected involvement of p97(VCP) translocase in the processing of heat-damaged proteins. Overall, we report an approach to inflict targeted thermal protein damage and its application to elucidate cellular stress-response pathways that are emerging as promising therapeutic targets.

[1] Laboratory of Genome Integrity, Institute of Molecular and Translational Medicine, Faculty of Medicine and Dentistry, Palacky University, Olomouc, Czech Republic. [2] Regional Centre for Applied Molecular Oncology, Masaryk Memorial Cancer Institute, Brno, Czech Republic. [3] Regional Centre of Advanced Technologies and Materials, Department of Physical Chemistry, Faculty of Science, Palacky University, Olomouc, Czech Republic. [4] Danish Cancer Society Research Center, Copenhagen, Denmark. [5] Division of Genome Biology, Department of Medical Biochemistry and Biophysics, Science for Life Laboratory, Karolinska Institute, Stockholm, Sweden. [6] These authors contributed equally: Martin Mistrik, Zdenek Skrott. ✉email: martin.mistrik@upol.cz; jb@cancer.dk

Exposure of cells and tissues to elevated temperatures is routinely used in research on protein thermal stability profiling, thermal therapies, treatments of accidental burns, and proteinopathies involving an accumulation of defective proteins. At the cellular level, the thermal damage primarily impairs proteins, causing their unfolding, aggregation, amyloidogenesis, and denaturation, phenomena particularly implicated in the pathobiology of Alzheimer's disease (AD), Huntington disease, Parkinson disease, amyotrophic lateral sclerosis and amyloidosis[1]. Accumulation of defective proteins is also among the hallmarks of cancer with potentially causative roles, and cellular mechanisms of protein quality control represent anticancer therapeutic targets, as exemplified by clinically applicable inhibitors of the ubiquitin–proteasome system (UPS)[2].

Studying responses to thermal damage of proteins on the level of a single living cell or even subcellular level represents a significant challenge due to the lack of available methods allowing precise and fast delivery of the heat to the target structure at the micrometer scale. The temperature elevation is currently achieved by heating the cell culture media using various heat sources such as a water bath, an incubator with a pre-set target temperature, or by various energy emitters, including microwaves, ultrasound, and infra-red lamps and lasers[3]. However, such approaches have severe limitations, including, but not limited to: (i) a significant time delay in achieving the desired temperature as the media must be heated primarily; (ii) spatial restrictions as the whole cultivation vessel, or at least its large part is inevitably exposed to the heat; and (iii) overall precision issues including the inability to target selected single cells or subcellular compartments, thereby limiting the type of biological questions one can address.

Currently, the function of cellular chaperones is studied mainly by biochemical approaches under in vitro non-physiological conditions, mostly based on model substrates and simplified peptides, leaving the role and regulation of chaperones under physiological conditions of intact cell largely unexplored. Dramatic differences between cells and test tube should be taken into consideration, such as molecular crowding (300–400 mg ml$^{-1}$ of proteins in cells), presence of other (macro)molecules, or increased interactions between macromolecules leading to changes in protein aggregation or folding[4]. Moreover, the findings from biochemical experiments are challenging to validate in cellular experiments due to a lack of appropriate methods. Thus, some basic questions regarding the function of chaperones in cells, including their recruitment kinetics to the substrates, remain unanswered.

The emerging field of plasmonic nanoparticles (NPs) has opened, besides other possibilities, a way for localized thermal therapy due to the efficient and tunable photothermal properties. When illuminated by light, free electrons localized on the NP surface become excited, and the local electron cloud is asymmetrically distributed over the whole NP. This distribution produces a coulombic restoring force between positively charged nuclei and negatively charged electrons from the conduction band, which leads to collective oscillation of the electron cloud on the particle surface called localized surface plasmon (LSP). The localized surface plasmon resonance (LSPR) occurs if the frequency of the incident light matches the frequency of LSP oscillation. As a result, the light is absorbed much more efficiently and generates localized and highly amplified electric fields in the proximity of the NP surface[5,6]. Absorption of light by NPs may be non-radiatively relaxed and simultaneously converted to heat energy. Surface plasmon resonance depends on many parameters of the NPs, including size, shape, composition, surface coatings, dielectric properties of the metal NP and the environment[6–8]. Importantly, the absorption and scattering frequency of plasmonic NPs can be selectively changed by adjusting the morphology and the structure of the NPs and tuned to be located in the desired wavelength. Silver NPs can be easily tailored to possess an intense SPR band at a suitable wavelength region, enabling them to produce heat after the irradiation with appropriate laser and makes them an excellent candidate for a photothermal therapeutic agent. Plasmon NPs convert energy from the light to heat immediately and efficiently, allowing localized heating of the surrounding environment[9–11].

In an attempt to remedy the lack of suitable techniques for inflicting targeted protein damage in live human cells, we exploit here the properties of the plasmonic nanosilver-modified surfaces as a cell culture substrate. The approach that we developed, and examples of its applications to study molecular pathways highly relevant for biomedicine, are presented below.

## Results

**Plasmon-coated cultivation surface as a tool for heat micro-irradiation.** We adopted the NPs technology to directly focus the heat on the individual cells or subcellular compartments within a micrometer scale. The method is based on modified microscopic cell culture plates, pre-coated by a layer of anisotropic silver NPs allowing excitation through targeted irradiation by conventional lasers used in the laser scanning microscopes (LSM) and allowing controllable heating. The deposition of NPs with suitable plasmonic properties on the cultivation surface is based on the layer-by-layer self-assembly technique, which facilitates the binding of negatively charged silver NPs using positively charged thin polymeric film deposited on the surface of the cultivation plate (Fig. 1a). For this purpose, water dispersion of anisotropic silver NPs of various crystallinity and shapes such as spheres, plates, rods, and triangles (Supplementary Fig. 1a, b), were synthesized by a two-step reduction method, showing the typical UV/VIS absorption spectrum (Supplementary Fig. 1c). Next, the prepared NPs were coated on the bottom of standard cell culture 24-well plates, which were pre-coated by a thin polymeric film consisting of polyacrylic acid (PAA) and poly(diallyl dimethylammonium chloride) (PDDA) polyions, thereby ensuring higher wettability and strong electrostatic binding ability of silver NPs, respectively[12].

Notably, such modified cultivation surface is chemically stable, optically transparent, and fully compatible with standard tissue culture methods, including cell adherence, cell viability and growth (Supplementary Fig. 1e–g). The photothermal effect and heat emission of the plasmonic modified cultivation surface after irradiation with the LSM laser is detectable within the LWIR spectrum (7.5–14 μm) by thermal imaging (Fig. 1b and Supplementary Fig. 1h).

To analyze the ability of the modified surface to induce microthermal damage of proteins in live cells, we employed a human reporter U-2-OS cell line expressing a GFP-tagged HSP70 protein (Heat shock protein 70). HSP70 is the central cellular chaperone involved in the processing of unfolded or aggregated proteins[4]. Immediately (within 8 s) after the laser exposure, HSP70-GFP accumulated at the micro-thermal damage sites, forming the laser path's precise pattern demonstrating proximate recognition of heat-damaged proteins by HSP70 in cells (Fig. 1c). The HSP70-GFP signal within the damaged areas changed over time positionally, and also the intensity decreased within a few minutes, indicating dynamic processing (Supplementary Video 1). HSP70 interacting partners, E3 ubiquitin ligase CHIP, and co-chaperone HOP[4] were also rapidly recruited to the sites of thermal damage with similar signal kinetics as HSP70 (Fig. 1c). Importantly, the same chaperone response was observed in another human reporter cell line (H1299) and on different plasmon layer-modified cell culture plates, thereby attesting to

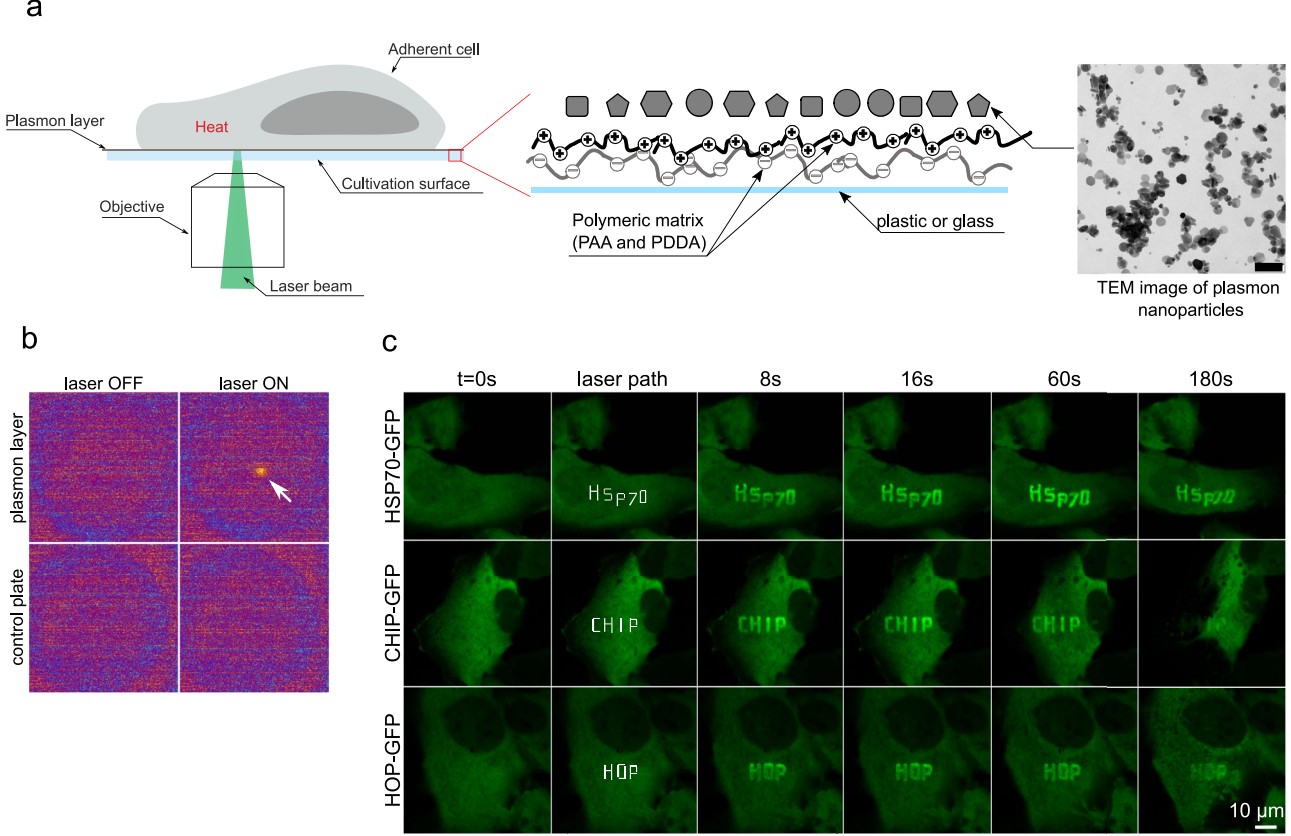

**Fig. 1 Plasmonic cultivation surface activated by a laser of appropriate wavelength induces microthermal damage. a** Schematic representation of the concept of microthermal damage inflicted on cellular proteins. The cell culture plate surface is modified by a thin polymeric film consisting of PAA (polyacrylic acid) and PDDA (poly(diallyl dimethylammonium chloride) for the efficient binding of plasmonic silver NPs. Plasmon NPs convert energy from light (laser) immediately and efficiently to heat, enabling direct focusing of the heat on subcellular regions. **b**. Thermal imaging shows emitted heat detected in the LWIR spectrum in plasmon-modified cell culture plate wells activated by 561 nm laser. **c** Recruitment of various GFP-tagged heat shock-related proteins to micro-heated regions in U-2-OS cells grown on a plasmon-modified Ibidi plate. Microheated regions were exposed to 561 nm laser (power 15%). The defined laser path is shown in white. Cells were followed in time. Representative results from two experiments. Scale bars = 10 μm, for TEM micrograph 100 nm.

the method's universal applicability (Supplementary Fig. 2a, b). Another chaperone involved in the processing of heat-damaged proteins, HSP90, was less prominent but also detectable within the damaged sites (Supplementary Fig. 2c, d).

To rule out the possibility that the observed protein damage and cellular response might also involve the direct damaging effect of plasmon activating laser, we performed an additional control experiment. The plasmon layer was partially scratched from the cultivation surface using a pipet tip, which is also visible using transmission light microscopy (Supplementary Fig. 3). Importantly, only those cellular areas which are in direct contact with the plasmon layer, but not those in contact with the adjacent scratched surface, revealed the typical HSP70 protein response upon exposure to the plasmon activating laser (Supplementary Fig. 3).

**Compatibility of the method with quantitative readouts**. Overall, the presented approach enables a so-far unprecedented analysis of the function and kinetics of chaperones or other factors involved in the processing of damaged proteins. In combination with a software-based ROI analysis, this setup also allows precise quantification of the process in time (Fig. 2a, b, and Supplementary Fig 4a). The microthermal damage can also be induced by an adaptation of the so-called 'laser stripe' (laser micro-irradiation) approach, which is commonly used in the field of DNA damage[13,14]. By this technique, dozens of cells can be simultaneously and uniformly exposed to co-linear laser stripes of damaged chromatin containing DNA double-strand breaks, forming an easily recognizable pattern[13]. We adapted this approach and the HSP70-GFP reporter for characterization of dose-response aspects of our method. Indeed, the dosing can be precisely controlled to trigger responses ranging from relatively faint and transient recruitment of HSP70-GFP to stripes, up to clearly visible stripes persisting for several minutes, by adjusting the laser power (Fig. 2c). Importantly, these experiments do not require any specialized laser equipment as the real energy hitting the plasmon layer corresponds to values 0.23, 0.38 and 0.48 mW, respectively. Alternatively, the total emitted heat can also be increased by changing the number of laser exposure cycles with a fixed laser power (Supplementary Fig. 4b). These data confirm that the method does not require any unusual equipment, and generates predictably reproducible results in a dose-dependent manner.

**Real-time kinetics and structural requirements of HSP70 recruitment**. As stated above, the current knowledge about chaperone function is based mainly on biochemical experiments with purified components under rather artificial conditions. HSP70 is known to form oligomeric structures, and recently, various oligomeric structures were proposed[15,16]. Yet, the relevance of HSP70 dimerization is not fully understood, and it has not been studied in the context of live cells.

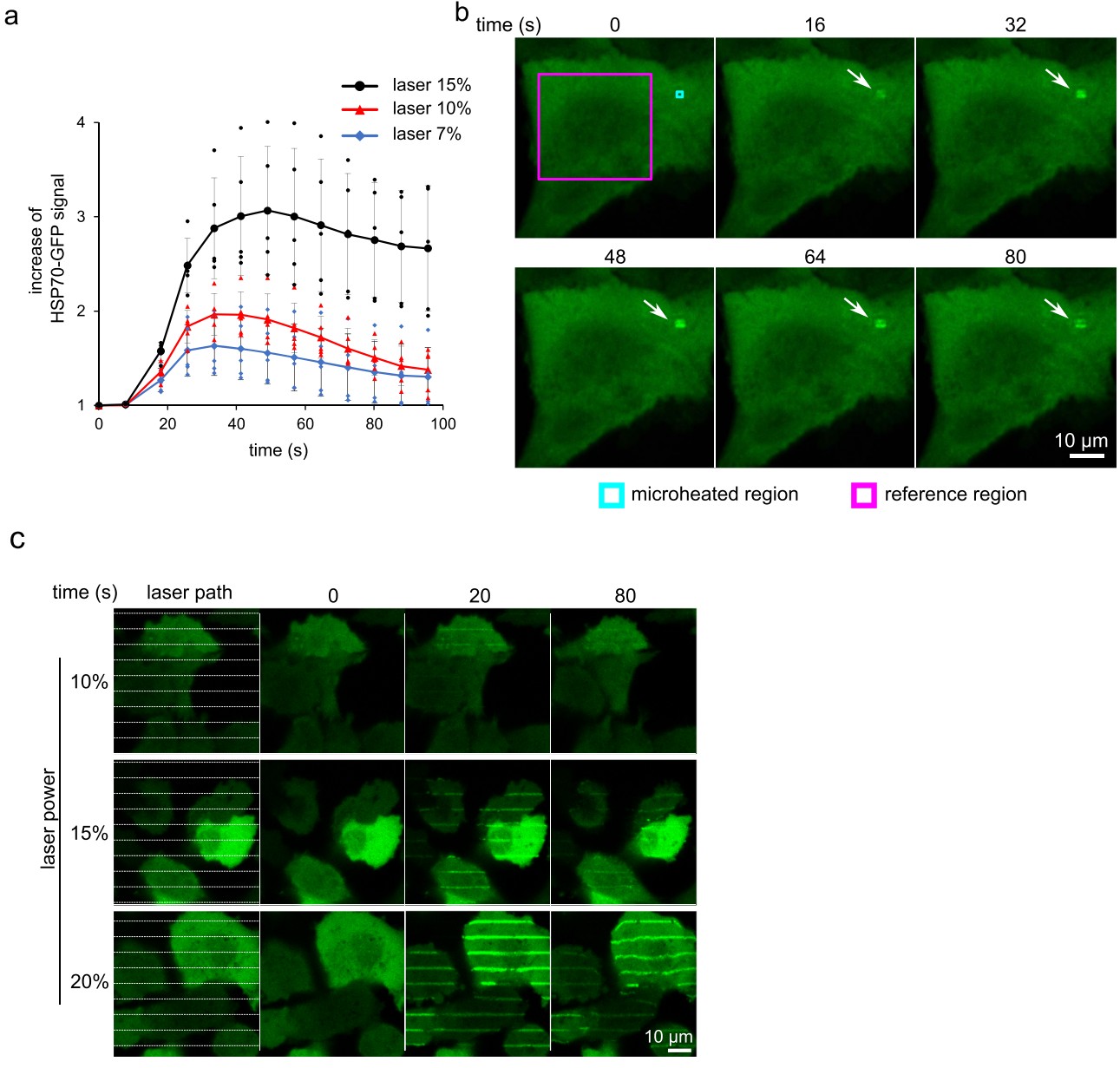

**Fig. 2 Demonstration of dose-response dependence and quantification of the cellular response. a** ROI-based quantitative analysis of the kinetics of HSP70-GFP recruitment to micro-heated proteins and the dependence of the process on the laser power used for plasmon layer activation (mean, SD, $n =$ 5 cells). **b** Representative images depicting the evolution of the HSP70-GFP signal in the micro-heated ROI (white arrow, $n =$ 5 cells). **c** H1299 cells grown on a plasmon-modified Ibidi plate expressing HSP70-GFP were exposed to collinear laser stripes of different laser power. The laser dose-response correlates with stronger and longer-persisting HSP70-GFP signals. Representative results from two experiments. Scale bars = 10 μm. Source data are provided as a Source Data file for Fig. 2a.

To assess the contribution of the HSP70 dimerization for its function, we employed three *HSP70* mutants known to impair the protein function in vitro[17,18], which have however so-far not been tested in cells. Using the thermal micro-irradiation approach, we observed that, compared to the positive control of wild-type HSP70-GFP, the mutation that impairs the HSP70's binding ability to client substrates (V438F) strongly abrogated the recruitment to micro-heated regions. This outcome was consistent with the expected mode of HSP70 recruitment to damage sites through recognition of the substrates, and it further attested to the suitability of our approach for addressing physiologically relevant questions. Also, the mutation affecting HSP70's ATPase activity (T204A), robustly impaired the recruitment of the respective mutant HSP70-GFPs

to the localized damaged proteins. Importantly, the subtle mutations (N540A, E543A) that impair the dimerization of HSP70 robustly inhibited the recruitment as well (Fig. 3a). These results reveal critical roles of the ATPase activity and dimerization, respectively, for proper recruitment of HSP70 to client substrates. Furthermore, these results further validate the applicability of our method for precision analyses of chaperone activity in the cellular context. These data also demonstrate that the observed recruitment to the lesion site does not reflect any potential unspecific method artefact but rather reflects HSP70's physiological ability to bind denatured proteins actively, in an acute manner, an on-demand dictated by the cellular context under heat-inflicted damage in live human cells.

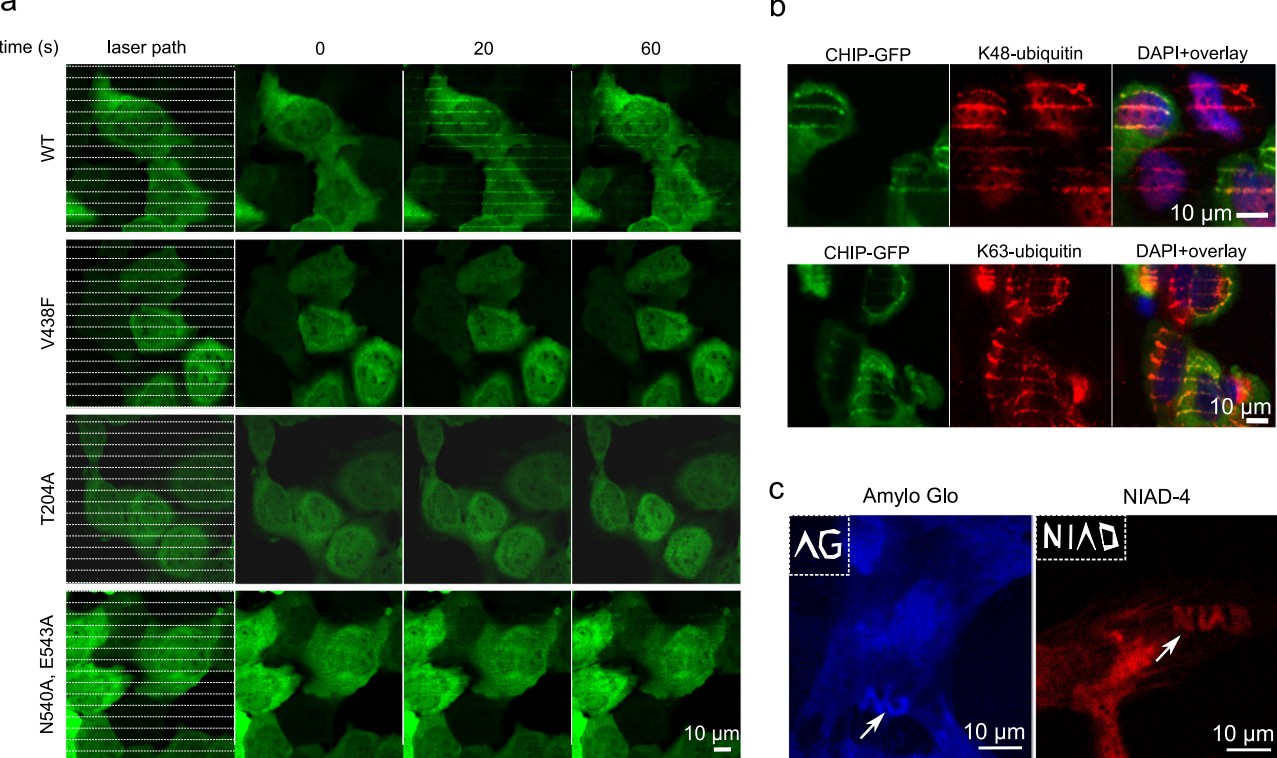

**Fig. 3 Impaired recruitment of several HSP70 mutants and characterization of proteins damaged by micro-heating. a** U-2-OS cells grown on a plasmon-modified Ibidi plate expressing HSP70-GFP variants were micro-heated by laser stripes and followed in time, revealing impaired recruitment to micro-heated regions of the indicated HSP70 mutants, compared to wild-type (WT) HSP70 control. **b** U-2-OS CHIP-GFP cells were micro-heated in the form of colinear stripes, fixed and processed for immunofluorescence analysis of the accumulation of K48- and K63-ubiquitinated proteins. **c** Detection of amyloid aggregates by NIAD-4 and AmyloGlo dyes in U-2-OS cells in microheated regions. The defined laser path is shown in the upper left corner of each image. All panels show representative results from two experiments. Scale bars = 10 μm.

**Thermally damaged proteins are ubiquitylated and processed by p97 translocase**. To provide further insights into the cellular response to micro-thermally damaged proteins and assess the method's compatibility with immunofluorescence, we next immunostained the exposed cells for ubiquitin. Using immunofluorescence, we observed a clear co-localization of the thermally damaged sites with signals from antibodies, specifically recognizing K48- and K63-ubiquitinated proteins (Fig. 3b). Such K48-linked ubiquitylation of the damaged protein indicates ongoing processing by the UPS, while the K63-ubiquitin is mainly associated with pathway signaling or autophagy[19], suggesting a potential involvement of additional mechanisms. Furthermore, we also tested specific fluorescent dyes recognizing unfolded or aggregated proteins. For example, NIAD-4 dye, commonly used as a detection reagent for β-sheet structures of AD-associated amyloid plaques, highlights the micro-heated proteins. Another dye accumulated within the thermally micro-irradiated regions is Amylo Glo (Fig. 3c), used for detecting amyloids. Thus, the amyloidogenesis of damaged proteins and the formation of β-sheets is induced within heated regions, consistent with previous publications reporting that heat stress triggers amyloid formation[20,21].

To elucidate the subsequent fate of the heat-damaged proteins, we considered their noticeable positional stability. It is well established that certain proteins dedicated to proteasomal degradation, which are part of insoluble cellular structures, require initial processing by p97 (VCP – Valosin Containing Protein, p97) as demonstrated for Endoplasmic reticulum-, chromatin-, or mitochondria-associated protein degradation[22]. Yet, any potential role of p97 in the processing of thermally damaged proteins has remained unexplored. To investigate whether p97 is involved in handling heat-damaged proteins, we first analysed the recruitment of GFP-tagged p97 to thermally damaged sites. Indeed, we observed the accumulation of p97-GFP within the micro heated areas a few minutes after irradiation and the persistence there for around 20 min (Fig. 4a). We further validated this response for endogenous p97 in human cells using immunofluorescence, and confirmed co-localisation of p97 with accumulated GFP-ubiquitin (Fig. 4b). For a more in-depth mechanistic insight, we pretreated the cells with the UAE1 (Ubiquitin-activation enzyme 1) inhibitor (MLN7243), which is capable of blocking nearly all cellular ubiquitinations[23]. Under the UAE1-inhibited conditions, we observed complete prevention of p97-GFP recruitment, indicating that ongoing ubiquitinations are required for the localisation of p97 within the heat-damaged sites (Fig. 4c). Interestingly, a specific inhibitor of ATPase activity of p97 (CB-5083)[24] also suppressed the recruitment of p97 to heat-damaged proteins (Fig. 4c), revealing that intact ATPase activity is required for proper accumulation of p97 within the heated subcellular regions.

To further study the active role of p97 in processing the heat-damaged proteins, we next studied a GFP-ubiquitin reporter cell line under p97 inhibition. In mock-treated cells, the GFP-ubiquitin was recruited to the micro heated regions within 5 min and persisted for up to 10 min. In contrast, cells pretreated with CB-5083 displayed stronger and longer-persisting GFP-ubiquitin signals within micro heated areas (at least for 20 min) (Fig. 4d). These data indicate direct and rate-limiting involvement of p97 in the proper processing kinetics of ubiquitinated proteins damaged by heat. We further confirmed these results under additional

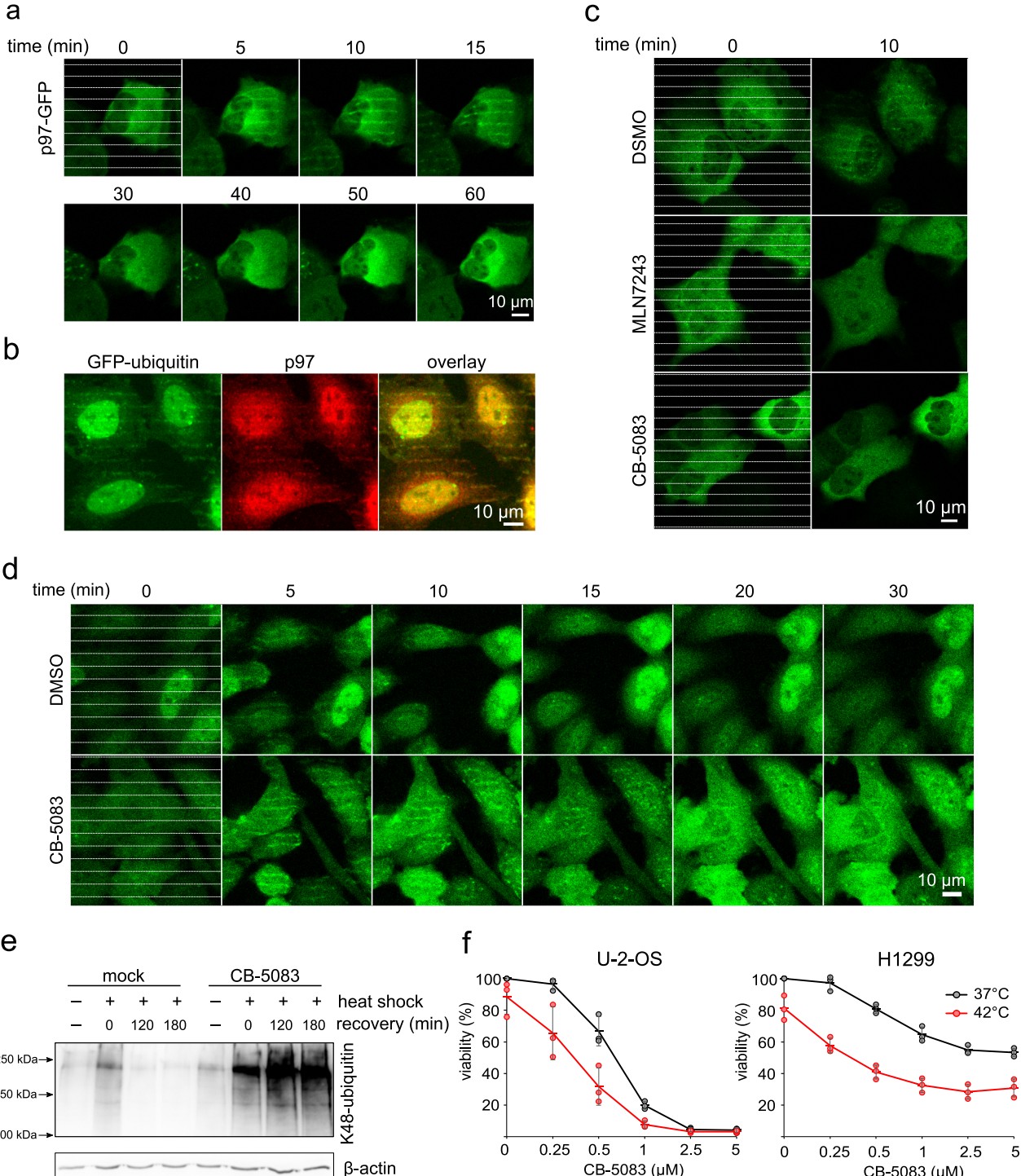

**Fig. 4 p97 is recruited to thermally damaged proteins. a** U-2-OS cells expressing p97-GFP grown on a plasmon-modified TPP plate were micro-heated by laser stripes and followed in time, revealing the kinetics of accumulation of p97-GFP in micro-heated regions. Representative results from three experiments. **b** U-2-OS cells expressing GFP-ubiquitin were micro-heated in the form of colinear stripes, fixed and processed for immunofluorescence analysis of the accumulation of endogenous p97 protein. Representative results from three experiments. **c** Pretreatment by UAE1 inhibitor MLN7243 (5 μM for 30 min) or p97 inhibitor CB-5083 (5 μM for 30 min) suppress recruitment of p97-GFP to micro-heated regions in the form of collinear stripes, which are visible only in mock-treated cells. Representative results from three experiments. **d** Inhibition of p97 by CB-5083 (5 μM for 30 min) increased the accumulation and persistence of GFP-ubiquitin in micro-heated regions in the form of collinear stripes compared to the mock-treated control in U-2-OS cells stably expressing GFP-ubiquitin. Representative results from three experiments. **e** p97 inhibitor CB-5083 (5 μM) increased the accumulation and persistence of K48-ubiquitinated proteins in insoluble cell fraction after heat shock (30 min at 43 °C). Cells were recovered at 37 °C for indicated times and cell pellets were analysed by WB. Representative results from two experiments. **f** Cell viability analysis after heat shock pulse (4 h at 42 °C) and recovery at 37 °C for 24 h (mean, SD from three independent experiments). Scale bars = 10 μm. Source data are provided as a Source Data file for Fig. 4e, f.

settings, based on a standard experimental approach used for the heat shock induction. We treated cells with CB-5083 or mock and then exposed the whole cell population to a 43 °C heat-shock pulse using a pre-warmed water bath, after which the cells were allowed to recover at 37 °C for different periods of time. Western blot analysis of insoluble cell fractions confirmed the involvement of p97 translocase in the processing of heat-damaged proteins. Indeed, CB-5083-mediated inhibition of p97 caused a more robust and persisting accumulation of K48-ubiquitinated proteins, that otherwise dynamically disappeared in mock-treated cells during recovery at 37 °C (Fig. 4e). Moreover, using a similar experimental setup, we confirmed that the cytotoxic effect of a heat shock pulse was significantly enhanced in cells with inhibited p97 (Fig. 4f), indicating a contribution of p97 to better survival of cells exposed to thermal insults.

**UPS compensates for the processing of thermally damaged proteins under HSP70 malfunction.** Our data from the recruitment dynamics of various responsive factors suggest two temporally distinct, and potentially linked or cooperating mechanisms involved in processing thermally damaged proteins. The initial, more acute mechanism involves immediate action of cellular chaperones and co-chaperones, represented primarily by HSP70, involving also HSP90, CHIP and HOP. The second, delayed and more durable mechanism, involves UPS, characterized by massive poly-ubiquitination and recruitment of the p97 translocase. To gain more insight into any potential orchestration within this two-wave cellular heat stress response, we evaluated the effect of HSP70 impairment by the established chemical inhibitor of HSP70, VER155008[21,25]. First, to validate the direct impact of VER155008 on HSP70 function in cells, we compared the HSP70 recruitment to damaged proteins in control and treated cells. VER155008 did not abrogate recruitment of HSP70 but rather resulted in prolonged HSP70 persistence at the heat damage sites, indicating inefficient chaperone-mediated processing of unfolded proteins, recognized, yet not further processed under HSP70 activity inhibitor treatment (Fig. 5a). To investigate the effect of HSP70 activity on the ubiquitination of damaged proteins, we then titrated damage intensity to the level at which the chaperones accomplished all the processing, i.e., without the apparent need for the subsequent involvement of p97. Under such settings, no GFP-ubiquitin signal was detectable. In contrast, under the same mild damaging conditions in cells with impaired HSP70 function (treated by VER155008), the accumulation of GFP-ubiquitin signal became detectable (Fig. 5b). Consistently, while under such mild conditions and proficient HSP70 response (mock treatment) p97-GFP did not recruit to sites of damage, in the VER155008-treated cells p97-GFP formed clearly visible stripes along the heat damaged subcellular areas (Fig. 5c). These results indicate that under a relatively mild heat damage conditions, the UPS pathway components including ubiquitin and p97, seem to provide a back-up compensatory role in case the primary chaperones (HSP70) are not fully operational. Furthermore, this two-wave mechanism becomes fully engaged, as a temporally coordinated cellular response, under conditions when the initial HSP-mediated pathway becomes overwhelmed by the severity of the damage.

## Discussion

Our present study describes a highly versatile method suitable for induction and monitoring of cellular responses to conditions that lead to unfolded, aggregated proteins and β-sheet amyloids, aspects highly relevant for both basic and translational research on cellular protein quality control and its malfunction in a range of neurodegenerative disorders and cancer. First, we designed and

validated plasmonic silver NPs modification of various cell cultivation microscopic plates. Such products enable the researchers thermal micro-irradiation of small subcellular regions and concomitant monitoring of both the overall fate, and particularly the heat-triggered intracellular events in adherently growing cells using standard LSM. Notably, the laser equipment required to apply this method does not demand any uncommon or highly specialized setups with regard to the laser power. Also, the wavelengths needed for the plasmon layer activation do not have to be strictly 561 nm as used in this study, given that the plasmon layer's absorption peak covers more laser types used in the diverse LSM-type laboratory microscopes.

In addition to the method itself, we applied this approach to study behavior of selected protein chaperones in the physiological context of live cells in a spatiotemporally-controlled manner and at the level of unprecedented detail. Indeed, the information about the substrate recruitment kinetics in real time, and the effect of some of the functional mutants of one of the most studied protein chaperons - HSP70 (including the requirement for intact substrate recognition, ATPase and dimerization domains, respectively) are now revealed owing to the method described here. We also aimed at obtaining more insights into the characteristics, and particularly the further processing of the thermally damaged proteins. One of the important contributions of our study to the field are the results revealing the recruitment kinetics and variable residence time of the heat shock factors at the damage sites in a heat dose-dependent manner. Furthermore, apart from confirming that heat damage induces protein β-sheet amyloidogenesis in cells, we now report that the heat-damaged proteins are not only recognized by specific chaperons but can be further modified by poly-ubiquitylation and processed by the p97/VCP translocase pathway. The latter two-phase scenario is valid for more severe damage or conditions of chaperone insufficiency. Surprisingly, both types of the poly-Ub chains (K48- and K63-linked) are present at the same time within the heat-damaged subcellular sites, implying that a coordinated action of multiple E3 ubiquitin ligases is to be expected, likely linked to further processing by different protein-maintenance pathways, an intriguing concept that should inspire further work in this area, now amenable for experimentation thanks to the technique we report.

From the available literature on yeast and bacteria, it is known that in the processing of cellular protein aggregates, HSP70 cooperates with AAA+ (ATPase associated with diverse cellular activities) family members such as HSP104 disaggregase. However, such disaggregase is apparently lacking in metazoans[26]. In this study using human cells, we discovered the involvement of the AAA+ translocase p97 in the processing of heat-damaged proteins. P97/VCP is an established component of the UPS machinery and generally protein quality control, promoting the degradation of ER-, mitochondria- or chromatin-associated proteins[22]. Despite mutations in the *p97* gene are associated with various neurodegenerative diseases accompanied by accumulation of protein aggregates, the function of p97 in processing heat-damaged and aggregated proteins has not been studied[27]. We show that p97 becomes recruited into the heat-damaged sites in the ubiquitin- and its own ATPase activity-dependent manner. The involvement of p97 becomes evident under more severe heat damage conditions, or in case the function of HSP70 is compromised (Fig. 6). After the p97 chemical blockade, we confirmed a substantial impact on the ubiquitin signal persistence within the heat-damage areas. Interestingly, the ubiquitin signal's disappearance was delayed, rather than completely blocked after p97 inhibition, suggesting that over time, spontaneous deubiquitylation of the damaged proteins or processing by alternative mechanisms such as autophagy or chaperone-mediated protein

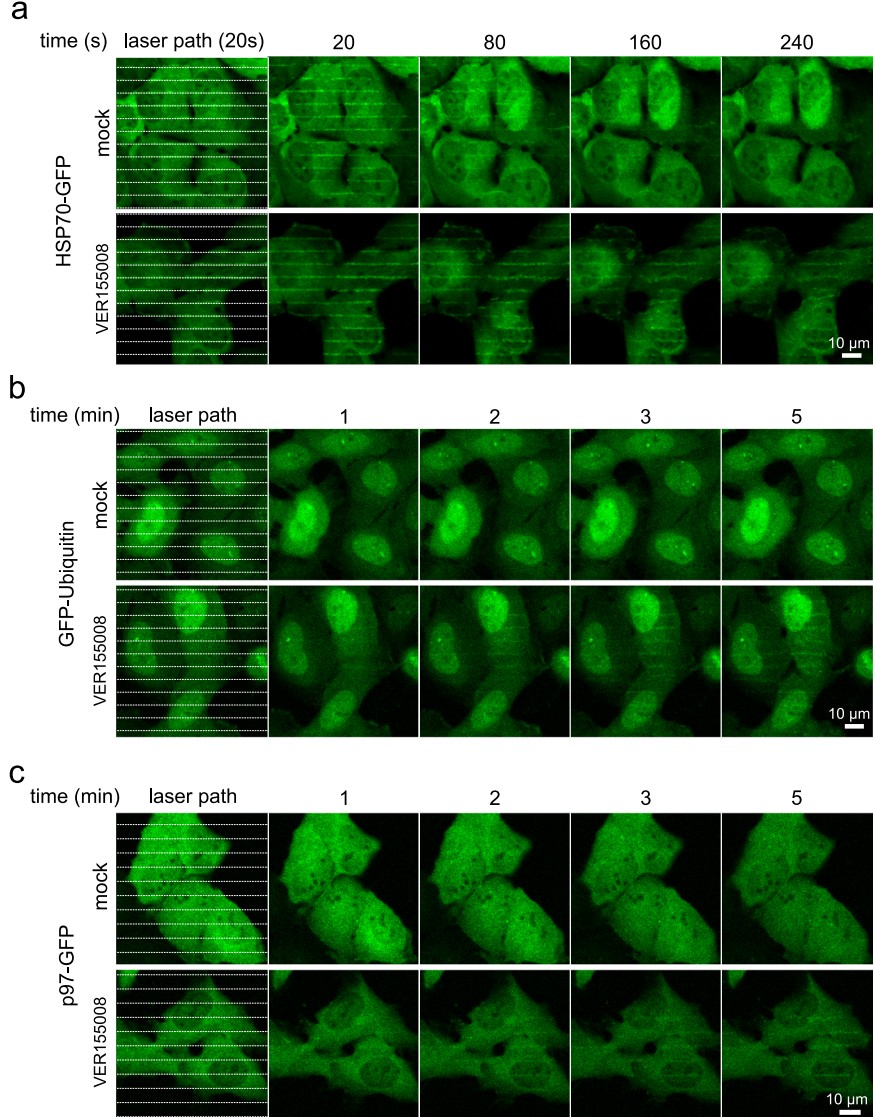

**Fig. 5 The effect of HSP70 inhibition on recruitment of HSP70, ubiquitin and p97 translocase. a** Pretreatment by HSP70 inhibitor (VER155008, 20 μM for 30 min) increases the persistence of HSP70-GFP in micro-heated regions. **b** Inhibition of HSP70 by VER155008 (20 μM for 30 min) promotes ubiquitination of heat-damaged proteins detected by GFP-ubiquitin reporter. **c** Inhibition of HSP70 by VER155008 (20 μM for 30 min) increases the recruitment of p97-GFP to micro-heated regions. All panels show representative results from three experiments.

repair may further join this complex cellular response. Broadly analogous with multiple-pathway involvement associated with vital cellular responses to insults such as DNA damage or oxidative stress[28,29], our current results further attest to the biological significance of a multifaceted cellular response to thermal damage. This emerging concept is further supported by our result reported here, that experimental inhibition of the ubiquitin/p97 arm of the response exacerbates cytotoxicity of the otherwise well-tolerated degree of thermal damage.

Overall, we present a versatile methodology that may be broadly applicable in life sciences, including diverse screening strategies to search for chemical or cellular modulators of chaperone function, and generally in both basic/mechanistic and translational biomedical research. Application of this approach allowed us to provide insights into the molecular mechanisms of cellular responses to thermal damage in real time, including temporal orchestration of complementary stress-response pathways. Last but not least, our study raises multiple questions that should inspire further research dedicated to the processing of damaged cellular proteins, an essential aspect of cellular biology

with broad implications for neurodegenerative, prion-associated, and other life-threatening diseases.

## Methods

**Synthesis of plasmonic NPs**. A step-by-step protocol describing the synthesis of plasmonic NPs can be found at Protocol Exchange[30]. Water dispersion of aniso-tropic silver NPs (108 mg/L) was synthesized by two-step reduction process, involving partial reduction of the $[Ag(NH_3)_2]^+$ complex cation by sodium bor-ohydride in the first step resulting in the formation of the silver nuclei, which were subsequently in the second step grown up by the reduction using weak reduction substance, e.g., hydrazine. All the reaction components were, at the laboratory temperature (23 °C), stirred continuously with a magnetic stirrer. Initially, 5 mL of aqueous silver nitrate (0.005 M), 1.25 mL of ammonia solution (0.1 M), 1.25 mL sodium citrate (1% w/w) and 13.425 mL distilled water were added into a 50 mL beaker and stirred while adding reducing agents. The reduction was initiated by the addition of 0.075 mL of sodium borohydride (0.001 M), which resulted in a reduction of silver complex cation, the formation of small NPs (seeds), and a change of the dispersion color to light yellow. Finally, 4 mL of hydrazine solution (0.05 M) was rapidly added into the dispersion of silver seeds under vigorous stirring, resulting in the growth of the seeds into final and stable silver anisotropic NPs followed by a change of the color of the dispersion from light yellow to typical purple. The final reaction concentrations of all the reaction components were as follows: silver nitrate $1 \times 10^{-3}$ mol dm$^{-3}$; ammonia $5 \times 10^{-3}$ mol dm$^{-3}$; citrate

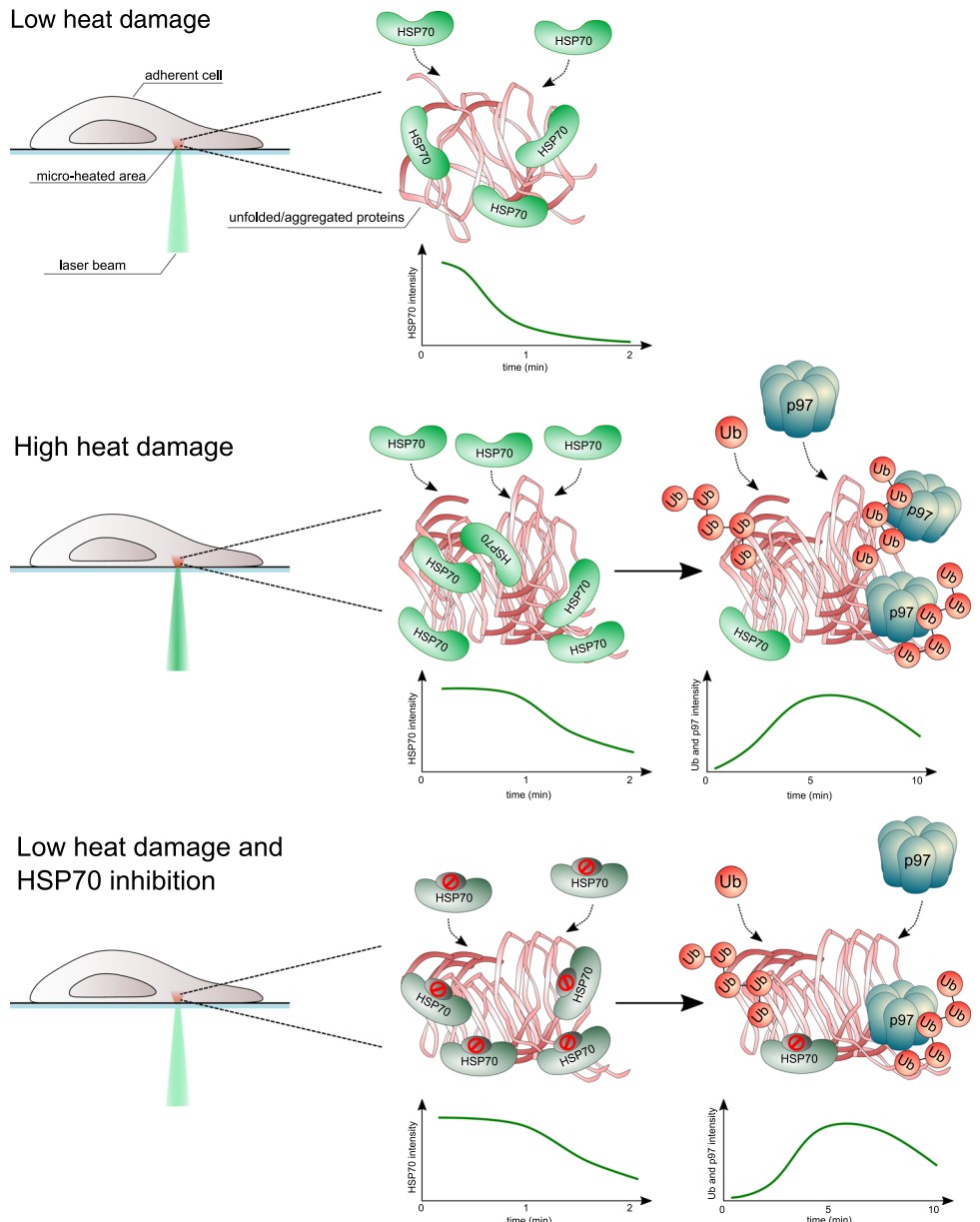

**Fig. 6 Schematic representation of the cellular two-wave response to micro-heat damage.** The intensity of micro-heat damage can be manipulated by laser power and defined regions enable detailed cellular response analysis. In case of low damage, the immediately recruited chaperones (such as HSP70, Heat shock protein 70) process the thermally damaged proteins within a minute and the activity of UPS is not needed. HSP70 persists for a longer time and ubiquitination (Ub) of damaged proteins is initiated in case of severe heat damage, followed by p97 recruitment and processing. In the case of HSP70 malfunction (inhibition), even low heat damage triggers ubiquitination and p97 recruitment as a compensatory response.

0.05% (w/w), sodium borohydride $3 \times 10^{-6}$ mol dm$^{-3}$ and hydrazine $8 \times 10^{-3}$ mol dm$^{-3}$ as a reducing agent. This way, prepared silver NPs were used for NP deposition on the cultivation surface. However, other plasmonic NPs with different shapes and optical properties (LSPR) can be prepared via the same method, but changing the ratio between citrate and hydrazine, using different amount of citrate varying from 0.25 to 4 ml (1% w/w) and adjusting the total volume to 25 ml. By this approach, the silver NPs with plasmon peak within the range of 440–725 nm can be synthesized. The prepared NPs have a negative surface charge ($\zeta = -38$ mV).

**Deposition of silver NP on cell culture plates with plastic surfaces**. A step-by-step protocol describing the deposition of silver NP can be found at Protocol Exchange[30]. Silver NPs prepared by the above-mentioned method were subsequently coated onto the bottom of functionalized Ibidi 24-well plates (u-Plate, Ibidi, cat.n.: 82406), Greiner CELLSTAR® 96-well plates (Sigma, cat. n.: M0562) or TPP 24-well plates (TPP, cat. n.: 92424). The plates were first functionalized by 1% PAA (Sigma Aldrich) solution followed by 1% poly(diallyldimethylammonium chloride) (PDDA) (Sigma Aldrich) solution for 2 h each to coat them by thin polymeric film consisting of two layers of oppositely charged polyelectrolytes

(PAA and PDDA). After 4 h of the treatment, wells were washed with distilled water to remove non-bonded polymers and filled with the dispersion of silver NPs. Silver NPs were bonded within 45 min to thin polymeric PAA_PDDA film deposited on well surface through the electrostatic interactions between negatively charged silver NPs and positively charged PDDA forming the top layer of the thin polymeric film. In the end, wells were washed with distilled water to remove unbonded NPs and air-dried.

**Deposition of silver NP on cell culture plates with glass surface**. A step-by-step protocol describing the deposition of silver NP can be found at Protocol Exchange[30]. Due to the fact that the glass surface is already negatively charged, the modification with negatively charged PAA as described above for plastic bottoms was skipped. Instead, the glass surface (Cellvis glass-bottom plates, P24-1.5H-N) was cleaned and activated by the piranha solution (H2SO4:H2O2, 7:3) for 15 min, followed by washing with distilled water. After that, the plates were functionalized by 1% poly(diallyldimethylammonium chloride) (PDDA) (Sigma Aldrich) solution for 2 h to coat them by thin polymeric PDDA film. After the treatment, wells were washed with distilled water to remove non-bonded polymer and filled with the

dispersion of silver NPs. Silver NPs were bonded within 45 min to thin polymeric PDDA film deposited on well surface through the electrostatic interactions between negatively charged silver NPs and positively charged PDDA forming the top layer of the thin polymeric film. In the end, wells were washed with distilled water to remove unbonded NPs and air-dried.

**Plasmids, cloning**. All coding sequences were cloned by Gateway recombination technology (Invitrogen, Carlsbad, CA, USA). The full coding sequences of human Hsp70 (HSPA1A, UniProt ID: P0DMV8-1), CHIP (STUB1, UniProt ID: Q9UNE7-1) and HOP (STIP1, UniProt ID: P31948-1) were cloned into PB-EF1a-N-EmGFP-PURO-GW-Dest vector containing an N-terminal GFP tag.Hsp70 point mutants were prepared by QuikChange Site-Directed Mutagenesis Kit (Agilent, Santa Clara, CA, USA) according to the manufacturer's manual. Following primers were used to create Hsp70 mutants: GACAACCAACCCGGGTTCCTGATCCAGGTGTAC and GTACACCTGGATCAGGAACCCGGGTTGGTTGTC for V438F, TGGGCGGGGGCGCCTTCGACGTG and CACGTCGAAGGCGCCCCCGCCCA for T204A and GTGTCAGCCAAGGCCGCCCTGGCGTCCTACGCCTTC and GAAGGCGTAGGACGCCAGGGCGGCCTTGGCTGACAC for N540A, E543A.

**Cell culture and transfection**. Human osteosarcoma U-2-OS and human lung cancer H1299 cell lines (both from ATCC) were used for all studies. Cells were maintained in DMEM media (Lonza) supplemented with 10% fetal bovine serum (Thermo Fisher Scientific) and 1% penicillin/streptomycin (Sigma-Aldrich). The Piggy Bac transposon system (PB) has been used to generate a stable expression of GFP fused proteins. A total of $10^5$ cells were transfected using Lipofectamine 3000 (Invitrogen, Carlsbad, CA, USA) with 1800 ng of transposon plasmid and 200 ng of transposase plasmid. Cells were selected in the same media supplemented with puromycin 5 μg/ml (InvivoGen, San Diego, CA). Flow cytometry and cell sorting (FACS Aria-III, Becton Dickinson) was used to enrich cell populations with optimal GFP expression. U-2-OS expressing GFP-ubiquitin reporter and U-2-OS expressing p97-GFP were previously described[31].

**Cell viability**. Cells were seeded on the original control multi-well plate (Greiner) or a plate modified by the plasmon NPs layer. Twenty-four hours after seeding, the cell viability was analyzed by XTT assay (Applichem) according to the manufacturer's instructions. XTT solution was added to the medium and incubated for 30–60 min, and then the dye intensity was measured at the 475 nm wavelength using a spectrometer (TECAN, Infinite M200PRO). To analyse the viability after heat shock, the cells were seeded on TPP 96-well plate. The next day, the cells were treated by CB-5083 or mock and placed in a water bath pre-heated to 42 °C for 4 h and then put back to cell culture incubator with standard conditions. Cell viability was analysed by XTT assay 24 h later.

**Microscopy and microthermal damage induction**. A step-by-step protocol describing the microthermal damage induction can be found at Protocol Exchange[30]. For the visualization and delivery of the microthermal damage, we used a Zeiss Axioimager Z.1 platform equipped with an LSM780 module for confocal laser scanning microscopy (CLSM). Used objectives included Zeiss objectives Alpha Plan-APOCHROMAT 40x water immersion for the Ibidi plates and glass-bottom plates, and LD Plan-NEOFLUAR 40x/0.6 Korr for the TPP plates. CLSM setup included argon 488 nm and 355 nm lasers for visualization. For the plasmon layer activation, we used exclusively 561 nm 20 mW solid-state laser. The power range hitting the plasmon layer corresponded to 0.16–2.41 mW; exact values are stated in the main text (see also details of laser power measurements below). The laser irradiation times were defined by the pixel dwell time and the total irradiation time. For the FRAP-like experiments where irradiation ROI was pre-defined, the pixel dwell time was fixed at 100 μs, and the whole irradiation time was dependent on the size of the ROI (~18–26 s for letter-like ROIs and 2 s for the square ROIs). For the striping approach, the pixel dwell time was fixed at 709 μs, and the total irradiation time was 0.85 s for one irradiation cycle resulting in 32 colinear stripes across the one microscopic field. See also ref. [13] for details for setting up the laser stripping approach in LSM. All laser irradiations and acquisitions were performed using the Zeiss Zen 11 software.

**Measurement of the laser power**. PT-9610 optometer with PD-2D Laser Power Detector (Gigahertz-Optik) was used to obtain the values describing the laser power in watts hitting the plasmon layer. The optometer was set for 561 nm wavelength, and the detector was placed instead of the microscopic plate. Irradiation of the sensor was performed by 561 nm solid-state laser via the same objectives as used for the micro-irradiation process. The values were recorded at continuous laser mode at different laser power setups. Exact values for Alpha Plan-APOCHROMAT 40× water immersion objective were: 0.16 mW (7% laser power), 0.23 mW (10% laser power), 0.38 mW (15% laser power), 0.48 mW (20% laser power), 2.41 mW (100% laser power). The exact value for LD Plan-NEOFLUAR 40x/0.6 Korr objective was: 1.27 mW (100% laser power).

**Quantitative ROI analysis**. The analysis was performed by the Zeiss Zen 11 software employing an internal plugin for bleaching and FRAP (Fluorescence recovery after photobleaching) analysis[31]. Control region (not exposed to the plasmon activating laser) was used for correction of baseline fluorescence and its potential change in time during acquisition and comparison with the microheated region (exposed to the plasmon activating laser). The data were exported to the Microsoft Excel 2016 and the increase of HSP70-GFP signal was calculated according to formula: increase of HSP70-GFP intensity =
$(\mathrm{mean}_{\mathrm{Microheated\ region}(t=x)}/\mathrm{mean}_{\mathrm{Reference\ region}(t=x)})/(\mathrm{mean}_{\mathrm{Microheated\ region}(t=0)}/\mathrm{mean}_{\mathrm{Reference\ region}(t=0)})$.

**Immunofluorescence and dye staining**. Cells were fixed 5 min after microthermal damage with 4% formaldehyde for 15 min at room temperature, washed with PBS, and permeabilized with 0.5% Triton X-100 in PBS for 5 min. After PBS washes, the cells on the plastic inserts were immunostained with primary antibody for 1 h at room temperature (anti-K48-ubiquitin, Apu2, Merck Millipore; anti-K63-ubiquitin, Apu3, Merck Millipore; anti-VCP, Abcam, ab11433), followed by PBS washes and staining with Alexa Fluor 568-conjugated secondary antibody for 60 min at room temperature. Nuclei were visualized by DAPI staining at room temperature for 2 min. For the beta-aggregates visualization, NIAD-4 (Sigma) dye was directly added to culture media (300 nM), and aggregates were detected in live cells after laser irradiation. AmyloGlo staining was performed according to the manufacturer's instruction (Biosensis). Briefly, cells were irradiated by the 561 nm laser, fixed with formaldehyde for 15 min at room temperature, washed with PBS, and permeabilized with 0.5% Triton X-100 in PBS for 5 min and washed. Next, the slide was incubated with 70% ethanol for 5 min, washed with distilled water, and stained with 1X AmyloGlo reagent for 15 min, followed by quick washes in 0.9% saline and distilled water.

**Thermal camera imaging**. The thermal camera (Therm-App®, Opgal Optronic Industries Ltd.) was placed in an in-house built holder keeping the camera at an ~5-cm distance from the top of the culture plate. The camera's germanium objective was manually focused on the bottom of a single well. The whole assembly was placed inside the Zeiss Axioimager Z.1 platform equipped with LSM780 module for confocal laser scanning microscopy (CLSM) (see Supplementary Fig. 1h for the setup). The bottom of the well's inner surface was focused and exposed to the 561 nm laser working in the continuous mode while the thermal camera was used to acquire thermograms.

**Western blotting**. U2OS cells were seeded at $0.7 \times 10^6$ cells per 6 cm dish for 24 h before the experiment. P97 inhibitor- (CB-5083; 5 μM; Selleckchem) or Mock-treated cells were exposed to 43 °C for 30 min in a water bath. Subsequently, dishes with cells were moved to an incubator with a standard set up of 37 °C and 5% $CO_2$. The cell lysates of pellet fraction were collected in time (0-, 2-,3 h) by a quick wash with 0,5% Triton X-100 followed by resuspension in 1× Laemmli sample buffer. Equal amounts of cell lysates were separated by SDS-PAGE on hand casted gels and then transferred onto a nitrocellulose membrane. The membrane was blocked in Tris-buffered saline containing 5% milk and 0.1% Tween 20 for 1 h at room temperature, and then incubated overnight at 4 °C with the following primary antibodies: anti-ubiquitin lys48-specific (1:1000; Merck Millipore, clone Apu2), anti-β-actin (1:1000; Santa Cruz Biotechnology, sc-47778), followed by detection with secondary antibodies: goat anti-mouse IgG–HRP (GE Healthcare), goat anti-rabbit (GE Healthcare). Bound secondary antibodies were visualized by ELC detection reagent (Thermo Fisher Scientific) and images were recorded by an imaging system equipped with a CCD camera (ChemiDoc, Image Lab 6.1 software, Bio-Rad).

**Characterization of plasmonic NPs**. The synthesized water dispersion of silver anisotropic NPs and the deposited layer of silver anisotropic NPs were characterized by transmission electron microscopy (TEM) using a JEM 2010 TEM instrument (Jeol, Japan). In the case of water dispersion, a droplet of the sample with a silver concentration of 108 mol dm$^{-3}$ was deposited on a carbon-coated copper grid and dried in a vacuum drier at 25 °C for 1 h. In the case of the silver NPs layer, carbon-coated copper grids were used and modified in the same way as cultivation plates. For this analysis, the carbon-coated copper grid was put in an Eppendorf tube and functionalized by PAA solution followed by poly(diallyldimethylammonium chloride) solution for 2 h each in order to coat them by the thin polymeric film. After 4 h of the treatment, carbon-coated grids were washed with distilled water in Eppendorf tube to remove non-bonded polymers and after that, Eppendorf tubes were filled with a dispersion of silver NPs. Silver NPs were bonded within 45 min to thin polymeric film deposited on carbon-coated copper grids and followed by washing with distilled water to remove un-bonded NPs and dried in a vacuum drier at 25 °C for 1 h. UV–vis spectra of silver NP dispersions were recorded on a Specord S 600 (Analytic Jena, Germany) spectrophotometer. Dispersions of silver anisotropic NPs were 10 times diluted prior to the measurements and layers of silver anisotropic NPs deposited on the cultivation plates were used as they were prepared. Zeta potential of silver NPs in water dispersion was obtained by electrophoretic mobility measurements using Zetasizer NanoZS (Malvern, UK).

**Reporting summary**. Further information on research design is available in the Nature Research Reporting Summary linked to this article.

## Data availability

The data that support the findings of this study are available from the corresponding author upon reasonable request. Source data are provided with this paper.

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

## Acknowledgements

The study was supported by MEYS CR: (Large RI Project LM2018129 - Czech-BioImaging), ERDF (project No. CZ.02.1.01./0.0/0.0./16_013/0001775), ENOCH project (No. CZ.02.1.01/0.0/0.0/16_019/0000868), and project CZ.02.1.01/0.0/0.0/16_019/0000754, Grant agency of the Czech Republic: GACR 20-28685S, GACR 19-03796S and GACR 19-22720S, Technology Agency of the Czech Republic: TN01000013, Ministry of Health of the Czech Republic (MMCI, 00209805), Danish Cancer Society (R204-A12617-B153), Danish Council for Independent Research (DFF-7016-00313), Danish National Research Foundation (DNRF125 - project CARD), Swedish Research Council (VR-MH 2014-46602-117891-30), and Lundbeck Fonden (R266-2017-4289).

## Author contributions

M.M., J.B., and Z.S. designed experiments, which were performed by M.M. and Z.S. A.P., L.H., and L.K. prepared and characterized plasmonic nanoparticle-modified surfaces. P.M. and V.V. prepared reporter cell lines. K.C. performed western blot analysis. T.B. performed cell viability tests. M.M., Z.S., P.M., A.P., and J.B. discussed and interpreted the results. M.M., Z.S., A.P., and J.B. wrote the manuscript, which was approved by all authors.

## Competing interests

M.M., Z.S., L.H., A.P., L.K., and J.B. are inventors of the pending patent application (European Patent Office, application number EP20198904.3; patent applicants: Palacky University in Olomouc, Czech Republic). The patent application covers preparation of plasmon-modified cultivation surfaces and usage of thereof for induction of micro-thermal damage. The remaining authors declare no competing interests.
