## [Peer Review File · Nature Communications]

Reviewers' Comments:

Reviewer #1:

Remarks to the Author:

The authors present a novel and interesting method of using silver nanoparticles to locally heat cells on the submicron scale and induce a thermal damage of proteins in the cell. A heat shock and denaturation of proteins was demonstrated on a few human cell lines. The method has multiple promising applications including localized hyperthermia. I recommend the manuscript for publication in Nature Communications after the authors address a few minor issues:

1. It would be beneficial for the readers to know the absolute power in watts, not only its percentage numbers indicated in Fig. 2 and in the text.
2. The authors should discuss the choice of 561 nm laser line given that the SPR peak of the colloidal silver nanoparticles is located at 420 nm and 580 nm as indicated in Fig. 1 Extended data.
3. A scale bar for the confocal microscope image in Fig. 1 c should be given in numbers.
4. The "surface charge" on page 7 line 191 should probably be changed to "zeta-potential".

Anatoliy Pinchuk

Reviewer #2:

Remarks to the Author:

Summary:

In this manuscript, the authors have established a method of thermal micro-irradiation of small subcellular regions in adherently growing cells. This method is based on modified microscopic cell culture plates, pre-coated by a layer of anisotropic silver nanoparticles (NPs) allowing excitation through targeted irradiation by conventional lasers used in the laser scanning microscopes and allowing controllable heating.

Specifically, the authors used plasmonic silver NPs because they produce heat after the irradiation with appropriate laser (references 8-10). The NPs with various crystallinity and shapes (Extended Data Fig. 1a,b) were then coated on the bottom of standard cell culture plates, which were pre-coated by thin polymeric film consisting of polyacrylic acid and poly(diallyl dimethylammonium chloride) polyions (Fig. 1a). It was revealed that human U2OS and H1299 cells adhere on the plasmon modified culture plate and are viable 24 hours after seeding (Extended Data Fig. 1e,f), and the heat emission after irradiation is detectable by thermal imaging (Fig. 1b, Extended Data Fig. 1g).

To test the ability of this method to induce microthermal damage in living cells, the author examined U2OS cells overexpressing HSP70-GFP fusion protein. They found that HSP70-GFP accumulated at the sites of microthermal damage after laser exposure, and thereafter the intensity decreased (Fig. 1c) within probably several minutes (Video 1). E3 ubiquitin ligase CHIP-GFP and HSP70 cochaperone HOP-GFP also accumulated at these sites with similar kinetics as HSP70-GFP in U2OS and H1299 cells (Fig. 1c, Extended Data Fig. 2a). Furthermore, thermal damage was induced by laser micro-irradiation, and dose-response effect of the laser irradiation was confirmed (Fig. 2a, Extended Data Fig. 2b).

To determine whether the accumulation of HSP70-GFP to the lesion site is associated with its function, the authors examined kinetics of three HSP70 mutants, V438F (cannot bind to client proteins), T204A (possesses dysregulated ATPase activity), and N540A/E543A (cannot dimerize). These mutants did not accumulate to the damage sites (Fig. 2b), suggesting that the accumulation

of HSP70-GFP is related with its chaperon activity. Furthermore, the authors confirmed the accumulation of K48- and K63- ubiquitin proteins (Immunofluorescence analysis) and amyloid proteins (Amylo Glo and NIAD-4 staining) to the thermally micro-irradiated regions (Fig. 3a,b). They conclude that "this method is suitable for induction and monitoring of unfolded, aggregated proteins and β -sheet amyloids" in the lesion sites.

General comments:

This study nicely applied plasmon silver NPs, which produce heat after the irradiation with appropriate laser, to the field of cellular heat shock response. The authors have established the method with high quality to induce microthermal damage in living cells, and showed suitability to monitor the accumulation of chaperones and co-chaperones as well as unfolded proteins and amyloids in the lesion sites, which are commonly observed in heat shocked cells or in the condensates of unfolded proteins. On the other hand, this study did not add anything about molecular mechanisms of the heat shock response and chaperone function. Specific comments are listed below.

Specific comments:

1) The authors state that "This method offers a unique opportunity to analyze heat shock response, chaperone functions, and the fate of unfolded/aggregated proteins ... (P5, line157-158)". In this manuscript, they must show at least some new mechanisms of the heat shock response, chaperone functions, or the fate of unfolded/aggregated proteins by using this method. The authors show in Fig. 2b that three mutants of HSP70-GFP did not accumulate in the thermally damaged sites. This could be a nice control to search new regulatory proteins for the accumulation of HSP70-GFP to and its dissociation from the lesion.

2) In Fig. 1b, is it possible to estimate the temperature of thermally micro-irradiated regions?

3) P3, line 93-94; "... fully compatible with standard tissue culture methods, including cell adherence, cell viability and growth". Data regarding growth on the modified culture dishes and morphology is important to monitor living cells. These data should be shown.

4) P4, line101; In Fig. 1c, the authors state that "Immediately (within a few seconds) after laser exposure". How long exactly the laser is exposed in Fig. 1c?

5) P4, lines 104-105; "the intensity decreased within a few minutes, indicating dynamic processing (Supplementary Video 1)". How do we know the intensity decreased within a few minutes in Supplementary Video 1?

6) P5, line 151; "... our method is suitable for in-cellulo induction". What does it mean?

Reviewer #3:

Remarks to the Author:

In this interesting manuscript, the authors designed and validated in biological experiments a new approach to inflict micro-thermal damage to subcellular areas. This promising methods, is based on localized surface plasmon resonance principle, adopted by the authors for direct focusing of the heat to the individual cells or subcellular compartments at a micrometer scale, a result currently not achievable by any state-of-the art alternative approaches. This new method may indeed find multifaceted applications in biomedicine, even well beyond the examples provided in the manuscript, and as such could really advance several fields of research in the future. While the method, as well as the interesting data on the heatshock cascade response are of wide interest and should be considered for publication, there are several points in which this dataset should be improved before publication can be recommended. Overall, the work is very innovative and promising, and as shown for some other techniques in recent past, there could be a potential for

many applications and hence also citations in the future, provided the authors are able to address the few weak points of the manuscript.

Criticisms and suggestions

1. In the abstract section, the authors claim that the subcellular thermal protein damage causes also some bystander effects. While this sounds intriguing, there is little evidence provided about this aspect of the method, and it would strengthen the manuscript if such data would be provided. Otherwise (a less favorable option), the relevant sentence in the abstract must be modified to remove such claim.

2. In Figure 1b, the authors show by thermal camera imaging, that heat is emitted only by plasmon-modified surface irradiated by laser. While useful another analogous specificity control must also be shown for the cell-based experiments describing the recruitment of heat shock-related proteins (e.g. HSP70). Microthermal protein damage in cells grown on plasmon-modified plates should be controlled with the same settings on control standard control plates, to exclude possible photo-damage of proteins directly by the laser.

3. In Figure 1c, the authors show the recruitment of various heat shock-related proteins, such as the HOP protein that serves as a co-chaperone linking HSP70 and HSP90 together. To further validate these new data and recruitment concept, it would be important to examine also the HSP90 itself, one of the most prominent cellular chaperones, which is currently missing in the manuscript.

4. To improve the overall impact of the message, and raise even a wider interest, the authors should extend the Discussion section, to elaborate on a several points for the benefit of the Nat Com broad readership. The authors should point out the current trend in biomedical research to study protein aggregation and its pathophysiological context, a broad area that would indeed benefit from implementing this new method. The authors should also better highlight the advantages of their approach over the current state of the art, and also better explain the biological findings of their heat shock experiments, particularly the mechanistic data with the HSP70 mutants.

Technical issue:

5. While the approach with modified plastic surfaces on its own is very valuable, I wonder whether the authors tried to also modify glass surfaces, and in such case what is their experience – since such modification could further extend the method's applicability.

Overall, we wish to point out that we found the comments and criticisms from the reviewers very constructive. By meticulously addressing the issues raised by the three reviewers, we have not only further specified some aspects of our new method, but we have also generated several pieces of additional mechanistic data about the cellular response to heat damage, that is now included in the revised manuscript. As a result, we have now also included a schematic model (in Fig. 6) of our emerging new mechanism of a two-wave molecular response to heat-damaged proteins, a novel concept that we hope will be of broad interest to researchers in several fields of biomedicine. It is our sincere hope that the revised version will now be approved by the reviewers and we would like to take this opportunity to thank the three reviewers for their stimulating comments which inspired a range of our new experiments and resulted in the ensuing strengthened dataset as presented in the revised manuscript.

REVIEWER COMMENTS

Reviewer #1 (Remarks to the Author):

The authors present a novel and interesting method of using silver nanoparticles to locally heat cells on the submicron scale and induce a thermal damage of proteins in the cell. A heat shock and denaturation of proteins was demonstrated on a few human cell lines. The method has multiple promising applications including localized hyperthermia. I recommend the manuscript for publication in Nature Communications after the authors address a few minor issues:

We thank the referee for the positive feedback and for pointing out some issues to be further improved. Those are addressed point by point below. We also appreciate the referees' non-anonymity.

1. It would be beneficial for the readers to know the absolute power in watts, not only its percentage numbers indicated in Fig. 2 and in the text.

We fully agree with the referee as this is an important piece of information for the research community, to know the laser power demands of our new approach. During the revision, we therefore measured the power of the plasmon layer activating laser by a specialized optometer. The sensor was placed under the microscope precisely in the position of the sample plate and irradiated using the same objectives as used for the experiments. The measured values are now included in the text's relevant passages (see text page 6), and the measurement procedure itself has been added to the Materials and methods part.

2. The authors should discuss the choice of 561 nm laser line given that the SPR peak of the colloidal silver nanoparticles is located at 420 nm and 580 nm as indicated in Fig. 1 Extended data.

This particular (561 nm) laser is the one available in our microscope and it fits within the range of the plasmon layer absorption peak. This type of green/yellow laser is among the commonly available laser sets used in the widely used laser scanning microscopes. In general, additional laser types with similar emission spectra would also be suitable for this application. Stimulated by this comment, we now mention this issue in the revised Discussion part.

3. A scale bar for the confocal microscope image in Fig. 1 c should be given in numbers.

As recommended by the Reviewer, we have now included the numerical values for all scale bars directly in the figure panels.

4. The “surface charge” on page 7 line 191 should probably be changed to “zeta-potential”.

Thank you - Yes, zeta potential is the correct term, and the text has been amended accordingly

Modified text: ‘The prepared nanoparticles have a negative zeta potential ($\zeta = -38$ mV).’

Reviewer #2 (Remarks to the Author):

Summary:

In this manuscript, the authors have established a method of thermal micro-irradiation of small subcellular regions in adherently growing cells. This method is based on modified microscopic cell culture plates, pre-coated by a layer of anisotropic silver nanoparticles (NPs) allowing excitation through targeted irradiation by conventional lasers used in the laser scanning microscopes and allowing controllable heating.

Specifically, the authors used plasmonic silver NPs because they produce heat after the irradiation with appropriate laser (references 8-10). The NPs with various crystallinity and shapes (Extended Data Fig. 1a,b) were then coated on the bottom of standard cell culture plates, which were pre-coated by thin polymeric film consisting of polyacrylic acid and poly(diallyl dimethylammonium chloride) polyions (Fig. 1a). It was revealed that human U2OS and H1299 cells adhere on the plasmon modified culture plate and are viable 24 hours after seeding (Extended Data Fig. 1e,f), and the heat emission after irradiation is detectable by thermal imaging (Fig. 1b, Extended Data Fig. 1g).

To test the ability of this method to induce microthermal damage in living cells, the author examined U2OS cells overexpressing HSP70-GFP fusion protein. They found that HSP70-GFP accumulated at the sites of microthermal damage after laser exposure, and thereafter the intensity decreased (Fig. 1c) within probably several minutes (Video 1). E3 ubiquitin ligase CHIP-GFP and HSP70 cochaperone HOP-GFP also accumulated at these sites with similar kinetics as HSP70-GFP in U2OS and H1299 cells (Fig. 1c, Extended Data Fig. 2a). Furthermore, thermal damage was induced by laser micro-irradiation, and dose-response effect of the laser irradiation was confirmed (Fig. 2a, Extended Data Fig. 2b).

To determine whether the accumulation of HSP70-GFP to the lesion site is associated with its function, the authors examined kinetics of three HSP70 mutants, V438F (cannot bind to client proteins), T204A (possesses dysregulated ATPase activity), and N540A/E543A (cannot dimerize). These mutants did not accumulate to the damage sites (Fig. 2b), suggesting that the accumulation of HSP70-GFP is related with its chaperon activity. Furthermore, the authors confirmed the accumulation of K48- and K63- ubiquitin proteins (Immunofluorescence analysis) and amyloid proteins (Amylo Glo and NIAD-4 staining) to the thermally micro-irradiated regions (Fig. 3a,b). They

conclude that “this method is suitable for induction and monitoring of unfolded, aggregated proteins and β -sheet amyloids” in the lesion sites.

General comments:

This study nicely applied plasmon silver NPs, which produce heat after the irradiation with appropriate laser, to the field of cellular heat shock response. The authors have established the method with high quality to induce microthermal damage in living cells, and showed suitability to monitor the accumulation of chaperones and co-chaperones as well as unfolded proteins and amyloids in the lesion sites, which are commonly observed in heat shocked cells or in the condensates of unfolded proteins. On the other hand, this study did not add anything about molecular mechanisms of the heat shock response and chaperone function. Specific comments are listed below.

Specific comments:

1) The authors state that “This method offers a unique opportunity to analyze heat shock response, chaperone functions, and the fate of unfolded/aggregated proteins ... (P5, line157-158)”. In this manuscript, they must show at least some new mechanisms of the heat shock response, chaperone functions, or the fate of unfolded/aggregated proteins by using this method. The authors show in Fig. 2b that three mutants of HSP70-GFP did not accumulate in the thermally damaged sites. This could be a nice control to search new regulatory proteins for the accumulation of HSP70-GFP to and its dissociation from the lesion.

We agree with the reviewer that mechanistic/biological insights into the heat shock response would significantly strengthen the manuscript, and therefore we devoted major effort to such work during the revision. Below, we list 4 major results, each providing a significant novel information that would advance the field, both conceptually, and mechanistically (some of this was already in the first version, but mostly these are newly added results, reflecting this stimulating comment from the reviewer):

1. We provide for the first time a precise real-time kinetics of Heat shock protein recruitment in live human cells. This challenging aspect has not been so far possible with the other available methods. Just to illustrate the significance of this advance, analogous studies of DNA damage factor recruitment to laser-inflicted DNA breaks were itself published in very high-impact journals...
2. Using a genetic approach (the mutants mentioned by the reviewer), we provide novel mechanistic information about which functional domains (and hence functional features) of HSP70 are required for the proper timely recruitment to thermal damaged sites. These genetic /functional requirements are manifest here in live human cells, and include the HSP70 dimerization and the ATPase activity as necessary prerequisites for the recruitment to the substrate. We agree that this novel information was not adequately highlighted in the original manuscript, which we have now tried to remedy in the revised version.
3. Based on newly performed sets of experiments, we now demonstrate in the revised version, that the response to thermal damage can occur in two temporally orchestrated waves, each involving a complementary pathway: the first acute wave by HSP70 and the associated chaperones, and the second, more delayed response, involving the ubiquitin-and p97/VCP translocase pathway, the latter as a novel information for the heat shock response research field.

4. We also show that this two-phase response is dose-dependent, and/or reflecting the proficiency (or malfunction) of the HSP phase, using different experimental manipulations to functionally modulate the capacity of either the HSP70 or the ubiquitin machinery or p97 translocase itself. Overall, the concept that is emerging is of a tightly coordinated, two-phase network, in which the recruitment and also residence time at the damage sites of the involved factors depends on the extent of the damage and the efficiency of the components of both pathways, with mutually compensatory roles (blocking/malfunction of one pathway boosts the recruitment/persistence of the other)- a completely novel aspect for the field of heat shock response research.

We are very grateful for this request, that allowed us to formulate the concept that is now outlined in the revised version, including a model in Figure 6

2) In Fig. 1b, is it possible to estimate the temperature of thermally micro-irradiated regions?

We did follow this suggestion by the reviewer, but as we explain below, so far this challenging task has proven to be technically unfeasible:

Thus, such analysis and determination of the temperature directly in thermally micro-irradiated regions' during the experiment itself in a confocal microscope cannot be technically performed due to several physical limitations:

- i) the limited space for placing a highly sensitive thermal camera in the body of the microscope;
- ii) Moreover, due to the resolution limitation resulting from the Abbe diffraction limit equation, even the best available germanium optics would not allow desired magnification to monitor such small ROI as used in the experiments. Such measurement would also need an extremely fast microbolometer chip enabling monitoring of the thermal changes in very short time periods.
- iii) Finally, even if the precise measurement would be possible, it could be done only in a dry environment as culture medium very efficiently blocks thermal wavelengths.

While not answering the raised question fully for the technical reasons, to illustrate our efforts in this direction, we show the experiment below, for the information of the Reviewer: We performed some additional measurements outside the microscope body using a high-quality thermal camera and the plasmon layer modified Ibody plate. An external laser with a wavelength of 532 nm and a power of 100 mW was used for this experiment. The laser spot was approximately 3 mm wide, and the experiment was performed at laboratory temperature. The conversion of absorbed light energy into thermal energy occurred immediately after irradiation of the silver nanoparticles layer when the temperature increased very shortly from 25.7 ° C to 35.7 ° C. This temperature was then constant throughout the irradiation. After the irradiation was finished (after 200 s), the silver nanoparticle layer's irradiated area was gradually cooled back to 25.7 ° C. The records of the temperature increasing during laser irradiation of Ibody plates with a thin layer of silver nanoparticles and without a layer of silver nanoparticles are shown in the Figure below.

Figure: Temperature recording using a thermal camera during laser irradiating (532 nm) the bottom surface of the Ibidi culture plate with a layer of silver nanoparticles (Ag +) and without a layer of silver nanoparticles (Ag-). (A) recording of the temperature increase (HEATING) of the bottom surface of the culture plate with the silver nanoparticle layer and (B) without the silver nanoparticle layer after starting the laser irradiation during the first 21 s and recording the cooling and temperature decrease (COOLING) of silver nanoparticle layer for 21 s from the end of the laser irradiation of the silver nanoparticle layer. (C) Graphical recording of the temperature increase and decrease after the start and the end of the laser irradiation of Ibidi plates with and without a layer of silver nanoparticles, respectively.

3) P3, line 93-94; "... fully compatible with standard tissue culture methods, including cell adherence, cell viability and growth". Data regarding growth on the modified culture dishes and morphology is important to monitor living cells. These data should be shown.

We agree with this comment and the relevant information is now included in the revised manuscript, particularly Supplementary Fig. 1f and 1g. The morphological analysis along with the comparable speed of growth and reaching confluency, as well as the cell viability, on both the standard tissue culture plastic and the plasmon layer modified surfaces are now presented side by side within the

context of the toxicity data (see Supplementary Figure 1f,g). In addition, cell morphology is also well apparent from the fluorescence microscopic images throughout the manuscript.

4) P4, line101; In Fig. 1c, the authors state that “Immediately (within a few seconds) after laser exposure”. How long exactly the laser is exposed in Fig. 1c?

The exact exposure times are now stated in the Material and methods under the chapter: Microscopy and microthermal damage induction

5) P4, lines 104-105; “the intensity decreased within a few minutes, indicating dynamic processing (Supplementary Video 1)”. How do we know the intensity decreased within a few minutes in Supplementary Video 1?

Our apologies for omitting to provide the timing in the original version of this video: The time scale in seconds is now enclosed in the revised Supplementary Video 1.

6) P5, line 151; “... our method is suitable for in-cellulo induction”. What does it mean?

We have eliminated this expression from the revised manuscript, and better described the experiments, to avoid any potential confusion.

(Just to clarify what was the original intended meaning of this expression: Some authors use “in-cellulo” to differentiate work done with cells grown in cell culture (i.e. live cells), from work using only cell extracts or purified proteins (i.e. from the strict ‘in vitro’ conditions). However, others insist that both live cell cultures and extracts/test-tube experiments should be referred to as in vitro experiments. While the latter “in vitro” expression does not really differentiate whether one is working with live cultured cells, dead cells, cell isolates, or just proteins or other cellular molecules; “in cellulo” stresses that you are working with living cells.)

Reviewer #3 (Remarks to the Author):

In this interesting manuscript, the authors designed and validated in biological experiments a new approach to inflict micro-thermal damage to subcellular areas. This promising methods, is based on localized surface plasmon resonance principle, adopted by the authors for direct focusing of the heat to the individual cells or subcellular compartments at a micrometer scale, a result currently not achievable by any state-of-the art alternative approaches. This new method may indeed find multifaceted applications in biomedicine, even well beyond the examples provided in the manuscript, and as such could really advance several fields of research in the future. While the method, as well as the interesting data on the heatshock cascade response are of wide interest and should be considered for publication, there are several points in which this dataset should be improved before publication can be recommended. Overall, the work is very innovative and promising, and as shown for some other techniques in recent past, there could be a potential for many applications and hence also citations in the future, provided the authors are able to address the few weak points of the manuscript.

Criticisms and suggestions

1. In the abstract section, the authors claim that the subcellular thermal protein damage causes also some bystander effects. While this sounds intriguing, there is little evidence provided about this

aspect of the method, and it would strengthen the manuscript if such data would be provided. Otherwise (a less favorable option), the relevant sentence in the abstract must be modified to remove such claim.

From the two options suggested by the Reviewer, we opted for the latter one and removed the note about the bystander effects from the abstract. The reason for this decision is not because the bystander effects are not reproducible (they are), but because we now provide a slightly evolved and mechanistically broader scope based on the comments of Reviewer no. 2, and consequently we now present a conceptually richer, mechanistically-oriented dataset, and the bystander effects are not critical for the model that we present, and they only reflected some observations which will need deeper research in the future.

Just to illustrate what kind of effect we see effect of thermally damaged cells on their ‘bystander’ cells. In the enclosed image below, you can see that a distant and relatively small part of a single cell’s body damaged by a robust heat insult can trigger prominent and speedy responses in surrounding cells, including cell death of cells not directly targeted by the laser. Broadly analogous bystander effects are known e.g. from the DNA damage field. For example, gamma-irradiated cells are known to trigger similar DNA damage effects and signaling patterns in neighboring cells that were not exposed (through complex cytokine signaling to surrounding cells). We will elucidate this aspect of the heat damage response in future publications.

Image depicting the bystander effect. Set of time-lapse images showing the evolution of the bystander effect in time. A red square marks the thermally irradiated area situated in a relatively distant part of one of the cells. The affected cell and also other cells in contact undergo quick morphological changes within seconds after irradiation. Green arrow points at a cell that was not thermally damaged but apart from morphological changes initiated also a quick cell death manifested by the cell body’s deterioration. For visualisation of the process, a reporter U-2-OS cell line expressing NPL4-GFP protein was used.

2. In Figure 1b, the authors show by thermal camera imaging, that heat is emitted only by plasmon-modified surface irradiated by laser. While useful another analogous specificity control must also be shown for the cell-based experiments describing the recruitment of heat shock-related proteins (e.g.

HSP70). Microthermal protein damage in cells grown on plasmon-modified plates should be controlled with the same settings on control standard control plates, to exclude possible photo-damage of proteins directly by the laser.

We fully agree with this comment and added such additional control, which we believe is even better than the referee's suggestion. In the plasmon layer coated plate, we removed the layer partially by scratching it off. The thermal cellular response is not detectable in the scratch areas, in contrast to the remaining coated areas. These data are now included in the revised manuscript (see Supplementary Fig.3).

3. In Figure 1c, the authors show the recruitment of various heat shock-related proteins, such as the HOP protein that serves as a co-chaperone linking HSP70 and HSP90 together. To further validate these new data and recruitment concept, it would be important to examine also the HSP90 itself, one of the most prominent cellular chaperones, which is currently missing in the manuscript.

We followed this suggestion, and new data for HSP90 recruitment are included in the revised manuscript (see Supplementary Fig.2d)

4. To improve the overall impact of the message, and raise even a wider interest, the authors should extend the Discussion section, to elaborate on a several points for the benefit of the Nat Com broad readership. The authors should point out the current trend in biomedical research to study protein aggregation and its pathophysiological context, a broad area that would indeed benefit from implementing this new method. The authors should also better highlight the advantages of their approach over the current state of the art, and also better explain the biological findings of their heat shock experiments, particularly the mechanistic data with the HSP70 mutants.

The manuscript has been modified to accommodate this stimulating suggestion, see the revised Discussion part.

Technical issue:

5. While the approach with modified plastic surfaces on its own is very valuable, I wonder whether the authors tried to also modify glass surfaces, and in such case what is their experience – since such modification could further extend the method's applicability.

This is another inspiring point. We did test also glass surfaces, and our method is also applicable for such experiments, after a little modification for glass surfaces. Data obtained with glass-bottom plates are now included (see Supplementary Fig. 2b), and the detailed protocol for such modification is now included in the Material and Methods part.

Reviewers' Comments:

Reviewer #1:

Remarks to the Author:

The authors addressed the comments given in the first review and now the manuscript can be published in Nature Communications.

Reviewer #2:

Remarks to the Author:

The authors extensively revised the manuscript, and now provide a new concept and mechanism of processing heat-damaged proteins, in addition to establishing a novel method to induce microthermal damage.

Reviewer #3:

Remarks to the Author:

The authors have thoroughly addressed my concerns and appear to have satisfactorily addressed the concerns of the other reviewers.